# Validation of the family focused mental health practice questionnaire in measuring health and social care professionals' family focused practice

Anne Grant[1]*, Susan Lagdon[2], John Devaney[3], Gavin Davidson[4], Joe Duffy[4], Oliver Perra[1]

1 School of Nursing and Midwifery, Medical Biology Centre, Queen's University Belfast, Belfast, Northern Ireland, United Kingdom, 2 School of Psychology, Ulster University, Coleraine, Northern Ireland, United Kingdom, 3 School of Social and Political Science, University of Edinburgh, Edinburgh, United Kingdom, 4 School of Social Sciences, Education and Social Work, Queen's University Belfast, Belfast, Northern Ireland, United Kingdom

* a.grant@qub.ac.uk

**Data Availability Statement:** All relevant data are within the paper.

## Abstract

### Background

Parental mental illness is a major public health issue and there is growing evidence that family focused practice can improve outcomes for parents and their families. However, few reliable and valid instruments measure mental health and social care professionals' family focused practice.

### Objectives

To explore the psychometric properties of the Family Focused Mental Health Practice Questionnaire in a population of health and social care professionals.

### Methods

Health and Social Care Professionals ($n = 836$) in Northern Ireland completed an adapted version of the Family Focused Mental Health Practice Questionnaire. Exploratory factor analysis was used to test the structure of the underlying dimensions in the questionnaire. The results, and theoretical considerations, guided construction of a model that could explain variation in respondents' items. This model was then validated using confirmatory factor analysis.

### Results

Exploratory factor analysis revealed that solutions including 12 to 16 factors provided a good fit to the data and indicated underlying factors that could be meaningfully interpreted in line with existing literature. From these exploratory analyses, we derived a model that included 14 factors and tested this model with Confirmatory Factor Analysis. The results suggested 12 factors that summarized 46 items that were most optimal in reflecting family

**Funding:** AG received an award from the NI Health and Social Care Board to complete this study (hscboard.hscbi.net or volorg.monitoring©hscni. net). The funders had no role in study design, data collection and analysis, decision to publish, or preparation of the manuscript.

**Competing interests:** The authors have declared that no competing interests exist.

focused behaviours and professional and organizational factors. The 12 dimensions identified were meaningful and consistent with substantive theories: furthermore, their inter-correlations were consistent with known professional and organizational processes known to promote or hinder family focused practice.

## Conclusion

This psychometric evaluation reveals that the scale provides a meaningful measure of professionals' family focused practice within adult mental health and children's services, and the factors that hinder and enable practice in this area. The findings, therefore, support the use of this measure to benchmark and further develop family focused practice in both adult mental health and children's services.

## Introduction

Internationally, it has been estimated that between 12 and 45% of adults receiving treatment from mental health services have children [1–3]. Reports from the United Kingdom (UK) suggest that 10% of mothers and 6% of fathers have mental health problems at any given time [4], with more current reports from the UK 'Understanding Society' survey [5] suggesting higher rates of 23.6% of mothers and 12.5% of fathers reporting symptoms of emotional distress. Northern Ireland (NI) is currently reported as having the highest levels of maternal mental illness within the UK [6], with one in four children aged 0–16 years exposed to maternal mental illness at any one time, and an estimated 53% of children over 16 having a mother who has been diagnosed with a common (i.e. depression and anxiety) or severe (i.e. psychosis) mental illness at some time [6]. Most recently, the Youth Wellbeing Prevalence Survey [7] found that one in five (22%) parents or caregivers across NI reported a previous diagnosis of any mental health disorder.

Parental mental illness (PMI) is an important global public health issue. The needs and issues for parents who have mental health problems, their children and their families are extensive and have been documented in numerous studies [8–14]. Bunting et al. found that parental mental health was one of the key factors shown to have a strong association with the development of mood and anxiety disorders among children and young people in NI. Children whose parents had current mental health problems (as measured by the General Health Questionnaire (GHQ-12)) were twice as likely to have an anxiety or depressive disorder themselves [7]. Internationally, it is estimated that 25 to 50% of children who have a parent with a mental illness will experience some psychological disorder during childhood or adolescence and 10–14% of these children will be diagnosed with a psychotic disorder at some point in their lives [15]. There is also substantial evidence that parental mental health problems may contribute to child maltreatment [15–17]. Moreover, stress from assuming a parenting role may negatively impact parents' wellbeing [18]. Having to juggle the demands of managing their own mental illness and additional responsibilities of managing their children's problems, can further heighten the risk of parents relapsing [17]. If parents perceive that they are unable to cope with their parenting role it may have a profound impact on their mood, self-esteem and self-efficacy [19].

A family focused approach to service delivery by professionals can help parents, children and other family members to prevent and/or cope effectively with the difficulties associated

with PMI [20, 21]. Within this approach, professionals engage the service user within the context of their immediate connected family relationships and endeavour to meet the needs of both service users and family members (including children) [22]. Central to this is being clear about how the needs of individual family members and the needs of the whole family are considered, especially when a parent's mental health may be impacting on their child to such a significant extent that there are concerns about protecting the child. A family focused approach may also involve directly engaging service users' children around issues related to PMI and promoting their capacity to understand and cope with it. Professionals may also indirectly support children by keeping them in mind while caring for service users, and by referral to other specialist support services as required. Activities can be classified as more or less family focused on a continuum, with direct support of service users' and their children (i.e. psychoeducation or family therapy) through to indirect support such as referral to other agencies. The types and intensity of activities and processes that professionals use to engage in family focused practice (FFP) are partly determined by the type of service in which professionals practice, their beliefs about the need for and importance of FFP, capacity to engage in it and how they think it should be operationalized [22].

In response to increasing recommendations by researchers, policy makers and professional organisations for health and social care professionals to engage in FFP [23–26], various countries, including the UK, have implemented organisational initiatives to promote FFP. For example, the Health and Social Care Board in NI has endorsed the use of The Family Model (TFM) [17, 27] as a framework to embed FFP within services and provide in-service training to increase awareness and use of the model in practice [28]. Assessment documentation in Northern Irish adult mental health and children's services has been refined in line with the domains of TFM [28], particularly domain one, where health professionals are prompted to identify children's needs related to PMI. Similarly, in other parts of the UK, TFM is the approach advocated for use by the Social Care Institute for Excellence [29].

## Theoretical foundation of the Family Focused Mental Health Practice Questionnaire

The Family Focused Mental Health Practice Questionnaire (FFMHPQ) [30] was developed against the backdrop of increasing recommendations for FFP; the broad variation in practice with families when a parent has a mental illness; and so the need for an agreed minimum skill set for FFP. The questionnaire was originally designed to measure mental health professionals' FFP working within adult services in Australia. Initially, Maybery et al developed the items for the FFMHPQ in parallel with a systematic review of the literature and with detailed input from the Victorian Families where a Parent has a Mental Illness (FaPMI) and the Co-coordinators and Vichamps project [31]. The review highlighted workplace policy and supports, worker skill and knowledge, and service user factors as central to FFP. These focal domains guided the initial generation of 100 items. These items were then subjected to rigorous review, rewrite and re- review, in Delphi focus groups with FaPMI coordinators. Subsequently, 14 subscales were developed, comprising a total of 45 items, which are scored on a seven-point Likert scale (ranging from strongly disagree = 1 to strongly agree = 7). Five subscales are said to measure professionals' family focused clinical practices and activities such as providing parenting support, referring family members to services and collaboration with other professionals to meet the needs of the family. The remaining subscales are said to measure organisational influences that can impact these activities such as workplace policies and support, and workload as well as issues related to service user engagement. Principal components analysis of the original scale in an Australian study [30], revealed a sixteen-factor solution, however, there was poor

internal consistency in three of the sixteen subscales. In addition, some of the remaining factors consisted of two items, indicating they may have been unstable [32].

Since its inception, the FFMHPQ has been adapted for use within a number of professional contexts internationally, particularly as the components of FFP are recognised and utilised across many sectors responding to issues associated with PMI. For example [33], adapted the measure for the early educational sector. In their study of preschool teachers and childcare professionals in Australia [33], removed eight of the original sixteen factors based on the research team's decision that they were not appropriate for teachers. Five of the remaining subscales possessed Cronbach's alpha values below 0.5, so consequently items were removed to improve reliability. Furthermore, in recognition of the need to acknowledge different language, culture and policy differences across countries, the FFMHPQ has been adapted for use in Ireland with mental health nurses [8] and translated into other languages including in Thailand [34] and in Norway [35, 36].

While none of these authors undertook psychometric evaluation, exploratory factor analysis (EFA) was undertaken as part of a study in NI with health visitors [37] and in Japan with mental health professionals [38]. While Leonard et al. [37] found a two-factor solution related to professional and organisational influences on FFP, Ueno et al. [38] found a 13-factor solution. Family focused practice is characterized by various activities on a continuum [20] and organizational and professional factors that enable and hinder it [8, 39]. The type of service and sector will influence professionals' understanding and interpretation of FFP and how and to what extent professionals use it in their practice [22]. Hence, the psychometric properties of the scale may vary in accordance with the particular profession and service context in which it is used depending on how FFP has been conceptualized, operationalized and embedded in practice.

Maybery et al. [30] recommended that future research using the FFMHPQ should examine and attempt to replicate the psychometric properties of the measure as well as attempt to expand its component structure to include additional important items and factors. Ueno et al. [38] also recommended that further research be conducted to examine FFP within different services, sectors and professionals to determine if there are differences that may impact FFP. Based on previous adaption and validation studies of the FFMHPQ, the aim of this study is to test the psychometric properties and factorial structure of the scale in a population of health and social care (HSC) professionals across NI. Our objective was to explore if family focused practices are influenced by both the individual professional and their core overlapping practices as well as the organizational structures which surround them. To our knowledge this is the first study that has evaluated the psychometric properties of the FFMHPQ when used within two distinct sectors, namely adult mental health and children's services. Results from this psychometric evaluation of the FFMHPQ can be used to inform future practice benchmarks and organizational developments.

## Methods

### Design

A cross sectional research design was implemented for the current study, with data collected using a survey approach. The survey consisted of three sections. Section one collected demographic data, section two included items from the FFMHPQ which is designed to measure professionals' FFP, and section three included items which aimed to capture, through a number of open-ended questions, professionals' experience of working with parents. Data pertaining to section two of the survey will be the focus of analysis within the current study.

## Measure

The FFMHPQ was developed by Maybery et al. [30] within the Australian context and further refined by Grant [39] within the Irish context. Within the current study, professionals responded to 21 subscales (which included 66 items) using a seven-point Likert Scale (ranging from strongly disagree to strongly agree). Subscales were designed to measure different family focused activities and behaviours (e.g. assessing the impact of PMI on the child, providing support to parents, carers and children), and factors that impact these (i.e. workplace support, policy and procedures, inter professional practice and professional development). A low score in a particular subscale suggests a reduced family focus practice in this area, with a high score suggesting increased family focus [30]. Psychometric information of the subscales is detailed in the works of Maybery et al. [30]. The measure was reported as having excellent content and construct validity and good internal subscale reliability.

As the FFMHPQ was devised for use in the Australian context, with a variety of professional disciplines (e.g. psychologists, psychiatric nurses, social workers), it required minor adaption and testing in the NI context particularly for health and social care (HSC) professionals practicing within adult mental health and children's services. Accordingly, the term 'consumer parent' was changed to 'service user' [8]. Further, in conjunction with developers of the original instrument, research team and advisory group, and in response to the emerging literature on PMI, FFP and organisational developments in FFP in NI, three additional subscales (containing 10 additional items) were included within the current survey. These new subscales aimed to measure professionals' understanding of The Family Model [17] and child protection protocols, and their interventions to reduce the impact of the parenting role on parental mental health. These new subscales are further detailed in Table 3 (Factor 4,7 & 9).

The validity of the FFMHPQ subscales outside the Australian adult mental health service context was also established. Initially an advisory panel assessed the items in the FFMHPQ subscales for their content validity. Panel members were selected for their expertise in FFP and PMI. All the items to be included were deemed relevant and therefore retained. The final survey including the FFMHPQ was then piloted in one organisation with ten HSC professionals (5 from children's services and 5 from adult services) not included in the main study to evaluate the clarity of the questions and their layout. The main changes made to the survey involved further refinement to the structure and language used particularly in relation to section three of the survey.

## Participants & procedure

Dissemination of the survey among HSC professionals across all five HSC Trusts in NI was achieved in two main ways, (1) online (via Survey Monkey) and (2) hard copy completion. Team managers across adult mental health and children's services were contacted via email and asked to circulate the information sheet and link to the online survey. The option for hard copy completion was also offered for those who preferred this method. Blank surveys were posted to requested teams using recorded delivery methods and later collected by a member of the research team. A database was created in SPSS for data entry of hard copy surveys; this database was later amalgamated with the online database of completed online surveys. Once both datasets had been merged, correct value range was checked. Additionally, every 25th hard copy survey was audited, and the data compared with the SPSS input data in order to ensure quality and consistency of manually entered data; these checks indicated that the data entry was reliable.

We sought to include a wide range of HSC Professionals working across adult and children's services with families where a parent has a mental illness. The survey respondents

broadly mirror the relevant workforce which has been the focus of Think Family NI initiatives. The survey was distributed to Approx. 3585 HSC professionals within adult mental health and children's services across the five HSC Trusts. The minimum number of HSC professionals needed to complete a survey ($n = 878$) was determined by various factors, including the size of the population to which results are generalizable to, the results of previous research and particularly findings from previous use of the FFMHPQ in different populations and countries and the overall purpose of the current study, which was to compare two groups of HSC professionals with regard to their FFP. Hence, a two sample comparison of means was used to estimate the overall sample size.

We ensured that the characteristics of respondents reflected the population of HSC professionals who fulfilled the inclusion criteria. To promote maximum variation and to secure sample access, a principal investigator (PI) for each Trust was identified along with an independent point of contact for the study.

A total of 1088 survey questionnaires were returned by HSC professionals giving a response rate of 30%. However, 119 of these were ineligible based on study inclusion criteria; 48 surveys completed by trainees and support workers, and 71 surveys completed by professionals in ineligible service areas (e.g. disability services) were excluded. Due to significant missing information, 101 cases were also removed from the dataset as more than 90% of the survey had not been completed and would not be suitable for inclusion in final analysis. The final sample comprised of 868 HSC professionals, a response rate of 24.2%. The total sample of HSC professionals was derived from all five HSC Trusts and included professionals from both adult mental health ($n = 493$) and children's social care services ($n = 316$) (Missing information regarding service area = 59). The largest number of responses were obtained from community mental health teams (28%), followed by children's services family intervention teams (18.1%), acute mental health and addictions inpatient services (9.3%), initial intake teams for children's services (Gateway Teams) (9.3%), community addictions teams (6.5%), 16yrs plus teams for adolescents (5.3%), crisis resolution home treatment (4.4%), and single point of access (0.9%). Given the variety of titles and terms attributed to different services across each Trust, the survey offered professionals the option to note their service area under a specialist mental health service or other category (15.2%). Such services included for example unscheduled care, Cognitive Behavioural Therapy (CBT), and those working within family centres.

A range of professional disciplines across these service areas participated. The most common were Social Workers ($n = 473$, 54.5%) followed by Nurses ($n = 293$, 33.8%). Other professionals, included Allied Health professionals ($n = 44$, 5.1%), Psychiatrists ($n = 33$, 3.8%), Psychologists ($n = 12$, 1.4%) and Other, for example, CBT Therapist ($n = 13$, 1.5).

Ethical approval was obtained from the Office for Research Ethics Committees Northern Ireland and Research Governance permission was obtained from the five HSC Trusts which provide statutory HSC services across NI (REC Reference 16/NI/0079). Participants gave informed consent; they were informed of the details of the study in online explanatory statements. Implied consent was obtained through participation in the completion of the online or hard copy, anonymous questionnaire.

## Statistical analysis

We used Exploratory Factor Analysis (EFA) to investigate the dimensions underlying the pattern of observed responses in the FFMHPQ. Exploratory Factor Analysis was run on the 66 items considering them as ordered categorical items. For this purpose, EFA parameters were estimated using a diagonally Weighted Least Square matrix with Missing Values (WLMSV) estimator, specifically designed for ordered data. We used a Geomin rotation, i.e. an oblique

rotation, which assumed correlations between the underlying dimensions. The use of the WLMSV estimator has the advantage of estimating parameters using all the information provided by respondents, even if information is incomplete (i.e. respondents do not answer all the questions). Approaches like this one, based on Full Information Maximum Likelihood (FIML), have been established to be more reliable and providing less biased results compared to traditional approaches that drop cases with missing data, as long as the reasons that can explain missing data can be assumed to be unrelated to the missing data themselves. Of note, 32 cases did not answer the items on which EFA analyses were based. Thus, results are based on *n* = 836 respondents that provided complete or incomplete data in the questionnaire.

The choice of the final models was based on the eigenvalues for the sample correlation matrix: factors were retained if the eigenvalue of factors was above 1. Furthermore, we estimated the model fit indices to check these provided adequate fit. These included the Comparative Fit Index (CFI) and the Tucker-Lewis Index (TLI): authors indicate values above 0.90 of these indices signal good model fit [40]. Other indices considered were the Root Mean Square of Error Approximation (RMSEA): a value <0.05 is considered to indicate a "close fit", while values <0.08 indicate adequate fit. Finally, the Standardized Root Mean Square Residual (SRMR) was also considered: values < 0.08 are considered to indicate adequate model fit. EFA was run using Mplus 7 [41].

The results of the EFA models were inspected and considered alongside theoretical considerations [9, 30, 38, 42] to develop a more parsimonious model that could explain the variation in participants' responses. The fit of this model was tested using Confirmatory Factor Analysis (CFA): CFA is a measurement model whereby the associations between observed items and underlying factors are explicitly modelled to allow items to load only on specific factors (i.e. factors are associated specifically with some items, but not with others). Associations between factors (or lack thereof) can also be modelled, alongside other parameters (e.g. associations between residual variances). For the sake of parsimony, when developing the measurement model through CFA, we retained fewer items in the models, taking into account results from the EFA and theoretical considerations in the selection of the items (i.e. retaining items that had substantive meaning according to our understanding of FFMHPQ).

The CFA was run considering the items as categorical ordered variables, similarly to our procedure in EFA. For this reason, we used a WLMSV estimator. Statistics considered in estimating model fit were, once again, the CFI, the RMSEA, and the SRMR. Models were estimated using Mplus 7 on all n = 835 participants who provided complete and incomplete responses to the items included in the final model.

## Results

### Exploratory Factor Analysis

Initial review of the scree plot suggested that Factors from 1 to 16 displayed eigenvalues above 1 therefore solutions with 16 factors or less were considered plausible solutions. As can be seen from Fig 1, a sudden shift between the 12 and 13 factor solution was evident, therefore the 12-factor solution was also considered as a plausible option and retained for further inspection.

Similarly, the solution with 14 factors was the first to provide adequate fit indices (including TLI) and was also retained for inspection. The fit of different solutions inspected are reported in Table 1.

After inspection of these solutions, we considered the solution with 14 factors as the solution that provided the best balance between statistical fit and interpretability of the solution based on evidence based literature [22, 30, 38, 43]. The model with 14 factors displayed

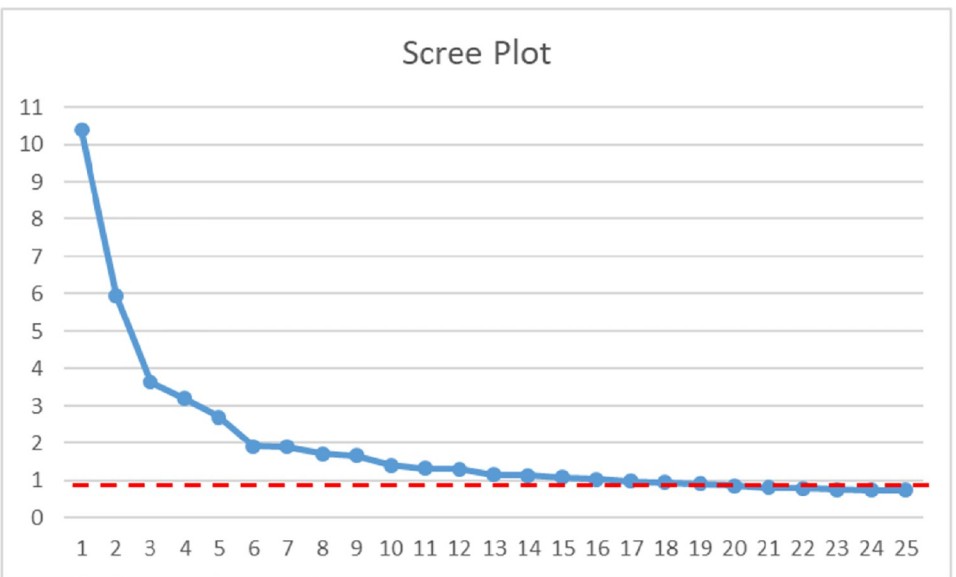

**Fig 1. Scree plot of Exploratory Factor Analysis.** The horizontal axis reports the factors from factor 1 to factor 25. The vertical axis reports the estimated eigenvalues associated with each factor. Eigenvalue = 1 is indicated by the dotted line.

adequate model fit, as reported in Table 1, and the item loadings provided meaningful and coherent interpretations of the underlying factors, which were in line with our substantive understanding of FFP. Factors seemed to represent some of the previously identified constructs of FFP such as skills and knowledge of professionals, workplace support and inter-professional practices [22]. Table 2 reports the items that were most strongly related with each factor.

Some correlations between the factors extracted were significant: these correlations are reported in S1 Table. Relationships between subscales measuring professionals' skills and knowledge (Factors 1, 2, 7, 8, 9), professional practice/ behavioural subscales (Factors 4, 5, 6), and higher order level subscales that describe organization/workplace factors that influence this (Factors 3, 10, 11, 12,) were observed.

## Confirmatory Factor Analysis (CFA)

The 14-factors model described in the previous section was the basis from which we derived a comprehensive more parsimonious model that included 12 factors. In what follows we

**Table 1. Fit indices of solutions 12 to 16.**

|  | 12-Factor Solution | 13-Factor Solution | 14-Factor Solution | 15-Factor Solution | 16-Factor Solution |
|---|---|---|---|---|---|
| Model parameters | 726 | 780 | 833 | 885 | 936 |
| CFI | 0.957 | 0.962 | 0.966 | 0.970 | 0.974 |
| TLI | 0.935 | 0.940 | 0.945 | 0.950 | 0.954 |
| RMSEA | 0.029 | 0.028 | 0.027 | 0.025 | 0.024 |
| SRMR | 0.028 | 0.026 | 0.025 | 0.023 | 0.022 |
| Chi-Square test of model fit, (degrees of freedom), and $p$ value | 2399.12 (1419); $p <$ .0001 | 2237.13 (1365); $p <$ .0001 | 2087.59 (1312); $p <$ .0001 | 1937.59 (1260); $p <$ .0001 | 1807.36 (1209); $p <$ .0001 |

**Table 2. Factor loadings by items and factors.** Loadings with * are significant at p = .05.

| No | Item | Factors | | | | | | | | | | | | | |
|----|------|---|---|---|---|---|---|---|---|---|---|---|---|---|---|
| | | 1 | 2 | 3 | 4 | 5 | 6 | 7 | 8 | 9 | 0 | 11 | 12 | 13 | 14 |
| 1 | My workplace provides mentoring to support health and social care professionals undertaking FFP | -0.044 | -0.106* | 0.657* | 0.057 | 0.072 | 0.003 | -0.011 | -0.008 | 0.203* | 0.012 | -0.02 | -0.004 | -0.096* | 0.093 |
| 4 | Government policy regarding FFP is very clear | -0.012 | 0.061 | 0.355* | -0.013 | -0.04 | 0.205* | -0.026 | -0.038 | 0.320* | -0.01 | -0.028 | -0.019 | -0.015 | 0.054 |
| 6 | I often receive support from co-workers in regard to FFP | 0.003 | 0.077 | 0.601* | 0.01 | -0.073 | 0.282* | -0.091* | -0.027 | 0.02 | 0.013 | -0.01 | -0.099 | 0.006 | -0.057 |
| 7 | I often receive support from co-workers in regard to FFP | 0.06 | -0.034 | 0.144* | -0.019 | 0.266* | 0.066 | -0.017 | 0.349* | 0.062 | -0.047 | 0.062 | -0.187* | -0.151* | 0.008 |
| 11 | I am able to determine the developmental progress of children whose parent(s) has mental illness | 0.046 | 0.107 | -0.053 | -0.006 | 0.001 | -0.068 | 0.119* | 0.511* | 0.267* | -0.054 | 0.063 | -0.013 | -0.179* | -0.09 |
| 12 | I sometimes wish that I was better able to help parents discuss the impact of their mental illness on their children | 0.412* | 0.082 | 0.035 | -0.038 | 0.104* | -0.116* | 0.002 | -0.164* | -0.01 | -0.005 | -0.067 | -0.186* | -0.158* | -0.117* |
| 13 | I am knowledgeable about how parental mental illness impacts on children. | -0.056 | 0.132* | 0.087* | 0.351* | -0.04 | -0.003 | -0.027 | 0.259* | 0.077 | 0.089* | 0.009 | 0.045 | -0.034 | -0.097* |
| 15 | I am able to determine the level of importance that parents who have mental illness place on their children maintaining attendance at day to day activities such as school and hobbies (e.g. sport, dance) | -0.061 | 0.549* | -0.074* | 0.136* | 0.055 | 0.029 | -0.015 | 0.076 | 0.263* | 0.08 | 0.068 | -0.001 | -0.064 | -0.059 |
| 17 | Working with other health and social care professionals enhances my FFP | 0.240* | 0.269* | 0.364* | 0.033 | 0.054 | 0.049 | 0.012 | 0.007 | 0.016 | 0.136* | 0 | -0.006 | -0.041 | 0.065 |
| 21 | At my workplace, policies and procedures for working with parents who have mental illness on family issues are very clear | -0.027 | 0.017 | 0.234* | 0.038 | 0.048 | 0.313* | 0.117* | -0.014 | 0.389* | 0.125* | 0.026 | 0.015 | 0.196* | 0.008 |
| 23 | In my workplace other workers encourage FFP | -0.028 | 0.163 | 0.623* | 0.017 | -0.031 | 0.212* | -0.031 | 0.004 | -0.082 | -0.002 | 0.021 | -0.250* | 0.099* | -0.021 |
| 24 | I provide written material (e.g. Think Family educational resources, leaflets) about parenting to parents who have mental illness | 0.029 | -0.041 | 0.236* | -0.077 | 0.169* | 0.439* | 0.137* | 0.197* | 0.024 | -0.02 | 0.006 | 0.05 | -0.012 | -0.065 |
| 28 | I am able to assess the level of children's involvement in their parent's symptoms | -0.05 | 0.290* | 0.038 | 0.093* | 0.054 | -0.012 | 0.041 | 0.339* | 0.305* | -0.154* | -0.042 | 0.006 | -0.063 | 0.023 |
| 29 | I should learn more about how to assist parents who have mental illness with their parenting | 0.615* | 0.167* | -0.079* | -0.023 | 0.072* | -0.041 | -0.086* | -0.011 | 0.022 | -0.027 | 0.006 | -0.006 | -0.081* | 0.05 |

(*Continued*)

**Table 2.** (Continued)

| No | Item | Factors | | | | | | | | | | | | | |
|----|------|---|---|---|---|---|---|---|---|---|---|----|----|----|----|
| | | 1 | 2 | 3 | 4 | 5 | 6 | 7 | 8 | 9 | 0 | 11 | 12 | 13 | 14 |
| 32 | I am able to determine the level of importance that parents who have mental illness place on their children maintaining strong relationships with other family members (e.g. other parent, siblings) | 0.076* | 0.618* | -0.059 | 0.047 | 0.007 | 0.139* | -0.001 | 0.102 | 0.232 | 0.014 | -0.012 | -0.017 | 0.08 | 0.019 |
| 33 | I refer parents who have mental illness to parent-related programs (e.g. parenting skills) | -0.021 | 0.106 | -0.037 | 0.163* | 0.696* | -0.049 | -0.021 | -0.015 | 0.133* | 0.042 | 0.05 | 0.007 | 0.026 | -0.002 |
| 34 | Children and families ultimately benefit if health and social care professionals work together to solve the family's problems | 0.433* | 0.280* | 0.076 | -0.03 | 0.019 | -0.059 | 0.094* | -0.092 | 0.002 | 0.190* | 0 | -0.037 | -0.019 | 0.224* |
| 35 | There is time to have regular contact with other agencies regarding parents, families or children (i.e. interface groups such as family support hubs) | -0.017 | 0.013 | 0.034 | 0.012 | 0.170* | -0.01 | -0.034 | -0.034 | 0.052 | 0.135* | 0.563* | -0.196* | 0.026 | -0.024 |
| 36 | I regularly provide information (including written materials) about mental health issues to children whose parent(s) has mental illness | 0.026 | -0.014 | 0.037 | -0.049 | 0.257* | 0.447* | 0.108* | 0.374* | 0.011 | -0.027 | 0.041 | -0.074 | -0.077 | 0.029 |
| 39 | I would like to undertake future training to increase my skills and knowledge for working with children whose parent(s) has mental illness | 0.853* | 0.008 | 0.01 | -0.057 | -0.001 | 0.02 | -0.026 | 0.236* | 0.028 | 0.066* | 0 | 0.011 | 0.04 | -0.006 |
| 42 | Team-working skills are essential for all health and social care professionals providing family-focused care | 0.435* | 0.230* | 0.064 | 0.046 | -0.082 | -0.007 | 0.023 | -0.082 | -0.079 | 0.261* | 0.023 | 0.044 | -0.053 | 0.128* |
| 43 | I often consider if referral to parent support programme (or similar) is required by parents who have mental illness | 0.157* | -0.017 | 0.018 | 0.219* | 0.484* | -0.026 | 0.057 | 0.048 | -0.008 | 0.019 | -0.045 | -0.02 | 0.04 | -0.042 |
| 44 | I would like to undertake training in future to increase my skills and knowledge about helping mentally ill parents with their parenting | 0.978* | -0.078* | -0.045* | 0.041 | -0.046 | 0.018 | -0.017 | 0.281* | 0.004 | 0.03 | -0.026 | -0.058* | 0.146* | -0.017 |
| 45 | I am skilled in working with parents who have mental illness in relation to maintaining the wellbeing and resilience of their children | -0.122* | -0.007 | -0.062 | 0.349* | 0.094* | 0.062 | -0.005 | 0.482* | 0.008 | 0.058 | 0.096* | -0.015 | 0.076 | -0.096* |
| 46 | I want to have a greater understanding of how to work within the multidisciplinary team to support children and families | 0.697* | -0.018 | -0.056 | 0.170* | 0.05 | 0.145* | -0.035 | 0.002 | -0.096* | -0.069 | 0.031 | -0.02 | -0.022 | -0.062 |

(Continued)

**Table 2.** (Continued)

| No | Item | Factors | | | | | | | | | | | | | |
|---|---|---|---|---|---|---|---|---|---|---|---|---|---|---|---|
| | | 1 | 2 | 3 | 4 | 5 | 6 | 7 | 8 | 9 | 0 | 11 | 12 | 13 | 14 |
| 47 | I provide education sessions for adult family members (e.g. about the illness, treatment) | -0.027 | 0.026 | -0.03 | 0.160* | -0.056 | 0.505* | 0.055 | -0.12 | -0.047 | 0.032 | -0.036 | 0.216 | 0.122 | -0.035 |
| 49 | I am knowledgeable about the key things that parents who have mental illness could do to maintain the wellbeing (and resilience) of their children | 0.01 | -0.058 | 0.023 | 0.593* | 0.008 | 0.021 | -0.061* | 0.361* | 0.043 | -0.002 | -0.082* | -0.06 | 0.076 | 0.101* |
| 50 | I am able to identify how parenthood can precipitate a parent's mental illness | 0.061* | 0.055 | 0.090* | 0.797* | -0.150* | -0.058 | 0.054 | 0.004 | 0.052 | -0.065* | -0.019 | 0.034 | 0.031 | -0.005 |
| 51 | I am able to identify how parenthood can influence a parent's mental illness | 0.051 | 0.045 | 0.033 | 0.871* | -0.084 | -0.078 | 0.029 | -0.074* | 0.01 | -0.013 | 0.03 | 0.023 | 0.017 | 0 |
| 52 | I assess the impact of the parenting role on the parent's mental health | -0.014 | 0.028 | 0 | 0.649* | 0.119 | 0.073 | 0.026 | 0.014 | -0.03 | 0.165* | -0.014 | 0.033 | -0.193* | 0.007 |
| 53 | I suggest practical strategies to facilitate parents who have mental illness to manage the dual demands of their parenting role and their mental illness or substance misuse | -0.028 | -0.008 | -0.051 | 0.529* | 0.119* | 0.158* | 0.016 | 0.034 | -0.147* | 0.229* | 0.017 | -0.005 | -0.205* | 0.056 |
| 54 | I understand how to use Falkov's Family Model to guide my FFP | -0.116* | -0.02 | -0.013 | 0.064 | 0.013 | 0.068 | 0.914* | 0.016 | 0.03 | 0.047 | 0.034 | -0.003 | -0.037 | -0.034 |
| 55 | I perceive that Falkov's Family Model can guide my FFP | 0.046 | 0.013 | -0.02 | 0.003 | 0.01 | 0.049 | 0.728* | -0.038 | 0.004 | 0.122* | 0.045 | 0.063 | -0.093* | 0.045 |
| 56 | I would need to undertake future training to increase my skills and knowledge for using Falkov's Family Model in practice | 0.528* | -0.024 | -0.018 | -0.004 | 0.004 | -0.031 | -0.381* | -0.008 | 0.021 | 0.258* | 0.023 | 0.037 | -0.113* | -0.014 |
| 57 | The regional child protection procedures are clear about when I should be concerned that a parent's mental illness is impacting negatively on a child | 0.03 | 0.023 | 0.091 | -0.017 | -0.121* | 0.127 | -0.039 | -0.012 | 0.289* | 0.534* | 0.008 | 0.002 | 0.140* | 0.007 |
| 59 | I discuss the impact of family functioning, on children's well-being, with the service user's adult family members/ carers | 0.017 | 0.075 | -0.034 | 0.139* | 0.105* | 0.043 | 0.015 | 0.284* | -0.087 | 0.310* | 0.072 | -0.212* | -0.084 | 0.071 |
| 60 | I would classify my interaction with children whose parent has mental illness as planned, purposeful involvement with therapeutic intervention | 0.03 | 0.008 | 0.015 | 0.096 | -0.043 | 0.131 | 0.066 | 0.446* | 0.002 | 0 | 0.264* | -0.250* | -0.074 | 0.018 |
| 54 | I understand how to use Falkov's Family Model to guide my FFP | 0.054 | -0.042 | 0.029 | 0.005 | 0.004 | -0.081 | 0.144* | 0.022 | 0.062 | 0.608* | -0.097 | -0.109 | 0.062 | -0.079 |

(*Continued*)

**Table 2.** (Continued)

| No | Item | Factors | | | | | | | | | | | | | |
|---|---|---|---|---|---|---|---|---|---|---|---|---|---|---|---|
| | | 1 | 2 | 3 | 4 | 5 | 6 | 7 | 8 | 9 | 0 | 11 | 12 | 13 | 14 |
| 66 | I know what to do if I was concerned that a parent's mental illness was having a significant negative effect on a child | 0.095* | 0.027 | -0.011 | 0.114* | -0.016 | -0.128 | 0.021 | 0.077 | 0.006 | 0.703* | -0.095 | 0.04 | 0.03 | -0.004 |
| 2R | In my area we lack services (e.g. other agencies) to refer children to in relation to their parent's mental illness (i.e. programs for children) | 0.024 | 0.023 | -0.027 | -0.004 | 0.006 | -0.003 | 0.099* | -0.02 | 0.042 | -0.093 | 0.152* | 0.226* | 0.486* | -0.027 |
| 3R | There is no time to work with children whose parent has mental illness or substance misuse around issues related to parental mental illness | 0.112* | -0.012 | 0.036 | -0.047 | 0.027 | -0.035 | 0.026 | 0.181* | -0.002 | -0.139* | 0.512* | 0.069 | 0.169* | 0.001 |
| 5R | Professional development regarding FFP is not encouraged at my workplace | 0.093* | -0.009 | 0.621* | 0.003 | -0.014 | -0.107 | 0.047 | 0.006 | -0.004 | -0.034 | 0.094* | 0.340* | -0.043 | -0.049 |
| 8R | I am not confident working with mentally ill parents on their parenting skills | -0.059 | 0.031 | 0.014 | 0.045 | 0.039 | 0.022 | -0.012 | 0.390* | -0.001 | 0.062 | 0.036 | 0.440* | -0.011 | -0.011 |
| 9R | I don't provide information to the carer and/or family about the service user's medication and/or treatment | -0.02 | 0.056 | 0.085* | 0.041 | 0.03 | 0.314* | 0.035 | -0.11 | -0.183* | 0.01 | -0.124* | 0.464* | 0.061 | 0.021 |
| 10R | Many parents who have mental illness do not consider their illness to be a problem for their children | -0.066 | -0.224* | -0.027 | 0.042 | -0.110* | 0.266* | -0.085 | 0.076 | 0.023 | -0.062 | 0.023 | 0.349* | -0.038 | 0.216* |
| 14R | There are no parent-related programs (e.g. parenting skills) to refer parents with mental illness to | -0.078 | -0.148* | 0.012 | 0.063 | 0.302* | 0.074 | -0.016 | -0.089 | 0.012 | 0.032 | 0.055 | 0.234* | 0.482* | 0.053 |
| 16R | I do not refer children whose parent has mental illness to child focused (e.g. peer support) programs (other than child and adolescent mental health) | 0.05 | 0.035 | 0.219* | -0.047 | 0.471* | -0.045 | 0.001 | 0.265* | -0.003 | -0.117* | -0.026 | 0.120* | -0.011 | -0.003 |
| 18R | My workplace does not provide mentoring to support health and social care professionals undertaking FFP | -0.059 | -0.124* | 0.706* | 0.004 | 0.062 | -0.026 | 0.037 | 0.044 | 0.009 | 0.052 | 0.059 | 0.042 | 0.055 | 0.051 |
| 19R | Due to location it is difficult to coordinate families and children with the required services | -0.031 | 0.138* | -0.013 | -0.102* | 0.033 | -0.006 | 0.075 | 0.037 | -0.03 | 0.048 | 0.198* | 0.003 | 0.361* | 0.001 |
| 20R | My workload is too high to do family focused work | -0.024 | -0.03 | 0.039 | 0.027 | -0.05 | 0.022 | 0.092* | -0.002 | -0.027 | -0.028 | 0.736* | -0.014 | 0.027 | 0.079 |
| 22R | My workplace provides little support for further training in FFP | -0.035 | -0.016 | 0.603* | -0.014 | 0.047 | 0.004 | 0.088* | 0.056 | -0.079* | 0.013 | 0.159* | 0.019 | 0.131* | -0.007 |
| 25R | I am not confident working with families of service user's | -0.068 | 0.143* | 0.041 | -0.130* | 0.025 | -0.032 | -0.027 | 0.374* | -0.096 | 0.187* | 0.024 | 0.227* | -0.032 | -0.038 |

(*Continued*)

**Table 2.** (*Continued*)

| No | Item | Factors | | | | | | | | | | | | | |
|---|---|---|---|---|---|---|---|---|---|---|---|---|---|---|---|
| | | 1 | 2 | 3 | 4 | 5 | 6 | 7 | 8 | 9 | 0 | 11 | 12 | 13 | 14 |
| 26R | Rarely do I advocate for the carers and/or family when communicating with other professionals regarding the service users' mental illness | 0.004 | 0.248* | 0.02 | 0.025 | 0.036 | 0.078 | -0.05 | 0.096 | -0.345* | 0.237* | 0.011 | 0.186* | 0.024 | 0.019 |
| 27R | Discussing issues for the service user with others (including family) would breach their confidentiality | 0.006 | 0.125 | 0.031 | -0.02 | -0.056 | 0.068 | 0.025 | 0.035 | -0.194* | -0.023 | 0.078 | 0.016 | -0.035 | 0.172* |
| 30R | I do not have the skills to work with parents who have mental illness about how parental mental illness impacts on children and families | -0.06 | 0.121* | 0.073* | 0.149* | -0.016 | -0.015 | -0.027 | 0.525* | -0.045 | 0.072 | 0.026 | 0.146* | 0.089* | 0.032 |
| 31R | There are no family therapy or family counselling services to refer parents who have mental illness and their children to | -0.044 | 0.015 | 0.04 | 0.023 | 0.251* | 0.006 | -0.081* | 0.025 | -0.051 | 0.013 | 0.094* | -0.035 | 0.587* | 0.016 |
| 37R | Rarely do I consider if referral to peer support program (or similar) is required by children whose parent(s) has mental illness | 0.044 | 0.120* | 0.093* | -0.06 | 0.389* | 0.039 | 0.113* | 0.298* | -0.076 | -0.104* | -0.111* | 0.023 | 0.046 | 0.081 |
| 38R | Children often do not want to engage with me about their parent's mental illness | 0.045 | 0.075 | -0.012 | -0.037 | 0.019 | -0.031 | 0.029 | 0.133* | -0.03 | -0.014 | 0.068 | 0.065 | 0.085 | 0.340* |
| 40R | I am not experienced in working with child issues associated with parental mental illness | -0.091* | 0.038 | -0.031 | -0.002 | 0.044 | -0.136 | 0.063* | 0.652* | -0.036 | 0.073 | -0.047 | -0.068 | 0.032 | 0.190* |
| 41R | I am not able to determine the level of importance that parents who have mental illness place on their children maintaining strong relationships with others outside the family (e.g. other children/peers, school) | -0.011 | 0.441* | 0.043 | 0.074 | -0.100* | -0.031 | -0.013 | 0.460* | 0.006 | -0.063 | -0.046 | 0.024 | 0.128* | 0.138* |
| 48R | I am not confident working with children whose parent(s) has mental illness | 0.016 | -0.065 | -0.024 | -0.014 | -0.023 | -0.194* | -0.01 | 0.635* | 0.007 | 0.082 | 0.009 | 0.077 | -0.058 | 0.105* |
| 58R | There is no time to work with families | 0.023 | 0.024 | 0.001 | 0.029 | 0.016 | 0.047 | -0.031 | -0.019 | -0.02 | -0.048 | 0.782* | 0.003 | 0.015 | 0.109* |
| 61R | Parents generally do not want to engage with me about the impact of their mental illness on their children | 0.019 | -0.025 | -0.018 | -0.02 | 0.02 | 0.033 | -0.016 | 0.049 | 0.186* | -0.03 | 0.057 | 0.229* | 0.029 | 0.657* |
| 62R | Discussing the impact of parental mental illness on children with parents who have mental illness would compromise rapport with them | -0.044 | 0.045 | 0.026 | 0.055 | 0.023 | -0.188* | 0.009 | 0.012 | -0.024 | 0.113 | 0.013 | -0.084 | -0.013 | 0.630* |

(*Continued*)

**Table 2.** (Continued)

| No | Item | Factors | | | | | | | | | | | | | |
|---|---|---|---|---|---|---|---|---|---|---|---|---|---|---|---|
| | | 1 | 2 | 3 | 4 | 5 | 6 | 7 | 8 | 9 | 0 | 11 | 12 | 13 | 14 |
| 63R | Insufficient numbers of health and social care professionals (i.e. nurse, social worker, clinical psychologist) in my service reduces worker's capacity to address parenting issues | -0.161* | -0.004 | 0.056 | -0.052 | -0.004 | -0.074 | -0.03 | 0.068 | 0.176* | 0.065 | 0.541* | 0.051 | 0.027 | 0.002 |
| 65R | I do not understand how to use Falkov's Family Model to guide my FFP | -0.093 | 0.008 | 0.070* | 0.01 | -0.018 | -0.08 | 0.702* | 0.054 | -0.006 | -0.073 | -0.063 | -0.026 | 0.104* | 0.017 |

describe the steps that led to this model. Items were retained based on recommended criteria that item loadings should be above .32, which represent over 10% of item variance explained by the underlying factor. Further, factors with fewer than three items or those items that cross loaded were assigned to the factor determined as a stronger associate with the exception of item 41R (*I am not able to determine the level of importance that parents who have mental illness place on their children maintaining strong relationships with others outside the family*) which was retained in factors 2 and 4, both of which address child related issues. We considered this as a more parsimonious solution that could adequately represent the variability in participants' responses particularly as factors 4 and 8 reflect distinctive knowledge regarding either the parent or the child which may also be linked to differences in professional focus and service remit (i.e. Adult mental health and Children services). The items retained in the analyses and the underlying factors, together with the factors' labels and their interpretations are reported in Table 3. In Table 4 we report the fit indices of this 12-factor solution. The model demonstrated adequate fit according to the CFI and the RMSEA. S2 Table displays the factor loadings of the items with the respective underlying factor.

**Table 3. 12 factor model including items and factor labels and definitions.**

| Factor | Items associated with factor | Label for Factors and Cronbach' alpha | Definition of Factors |
|---|---|---|---|
| 1 | 29, 39, 44, 46, 56 | Training (.83) | Professional willing to undertake further training |
| 2 | 15, 32, 41R | Connectedness (.67) | Professionals capacity to assess parent awareness of child connectedness |
| 3 | 1, 5R, 6, 18R, 23, 22R | Workplace support (.81) | The workplace provides support (e.g. supervision and professional development for family focused practice). |
| 4 | 49, 50, 51, 52, 53 | Skill and Knowledge to support parent and their parenting (.79) | Professional's knowledge and skill regarding impact of parenting on parent's mental health and activities supporting parent in parenting role |
| 5 | 33, 37R, 43 | Referral (.60) | Referral to other services |
| 6 | 24, 36, 47 | Psycho education (.48) | Provision of psycho education for parents, children and other adult family members |
| 7 | 54, 55, 65R | Understanding The Family Model (.78) | Awareness of TFM and its relevance in practice |
| 8 | 11, 30R, 40R, 41R, 48R | Skills and knowledge to support children (.75) | Professional's knowledge and skill to support children directly and via the parent |
| 9 | 57, 64, 66 | Child protection (.49) | Professional's understanding of child protection protocol |
| 10 | 3R, 20R, 35, 58R, 63R | Time and workload (.77) | Time or workload issues regarding family focused practice |
| 11 | 2R, 14R, 31R | Service availability (.64) | Lack of services to refer parents and children to |
| 12 | 38R, 61R, 62R | Engagement issues (.60) | The opportunity for engagement with parents and children about PMI |

Table 4. Model fit parameters of the 12-factor solution used in CFA.

| | 12-Factor Solution |
|---|---|
| Model parameters | 389 |
| CFI | 0.90 |
| TLI | 0.89 |
| RMSEA | 0.049 |
| WRMR | 1.573 |
| Chi-Square test of model fit, (degrees of freedom), and *p* value | 2734.43 (922); *p* < .0001 |

All the items included in the model had generally strong relationships with the underlying factor and were significant at *p* < .001., as reported in S3 Table. Furthermore, the factors showed patterns of correlations that were consistent with substantive knowledge and theories. These correlations are reported in Table 5. Note that some correlations were negative and significant (e.g. correlations between Factor 1 *Training* and Factor 7 *Understanding The Family Model* = -0.322; Factor 1 *Training* and Factor 11 *Service Availability* = -0.321). We suggest that this may be reflecting a relationship between those indicating a need/willingness for further family focused training and those who indicated they had less understanding of The Family Model and that those who felt they needed more training also perceived that there was a lack of services to refer parents and children to.

A high number of remaining relationships were positive and strong (r >.32), with 10% of shared variance between these factors. Subscales measuring professionals' skills and knowledge were positively correlated across one another (e.g. *Skills and knowledge to support children* WITH *Connectedness; Child Protection* WITH *connectedness*). Professional practice/ behavioural subscales were also positively correlated with these individual level subscales (e.g. *Skills and knowledge to support parent with their parenting* WITH *Connectedness*; *Referral* WITH *Connectedness*; *Understanding The Family Model* WITH *Psycho Education and Referral*; *Skills and knowledge to support children* WITH *Skill and Knowledge to support parent and their parenting*, *Referral* and *Psycho education*; *Child protection* WITH *Skill and Knowledge to support parent and their parenting*) as well as across one another (e.g. *Referral* WITH *Skill and Knowledge to support parent and their parenting*; *Psycho education* WITH *Referral*).

Higher order level subscales that describe organisation/workplace factors were also positively correlated with individual professionals' skills and knowledge factors (*Understanding The Family Model* WITH *Workplace support*; *Engagement issues* WITH *Skills and knowledge to support children; Time and workload* WITH *Skills and knowledge to support children*) and professional practice/ behavioural factors (e.g. *Referral* WITH *Workplace support*; *Psycho education* WITH *Workplace support*; *Time and workload* WITH *Psycho education*) as well as across one another (Time and Workload WITH Workplace support, *Service availability* WITH *Workplace support* and *Time and Workload*; *Engagement issues* WITH *Time and Workload*). These findings are reflective of proposed models of family focused practice which suggest that specific family focus practises are influenced by both the individual professional and their core overlapping practices as well as the organisational structures which surround them [20, 22, 44, 45].

## Discussion

The current study set out to test the psychometric properties and factorial structure of the FFMHPQ in a population of Northern Ireland HSC professionals working within adult mental health and children's services. Our objective was to explore if family focused practices are

**Table 5. Correlations between factors of 12 factor model.**

| | Correlations | Std Error | *p* value |
|---|---|---|---|
| Factor #2 Connectedness WITH | | | |
| Training | 0.161*** | 0.042 | < .001 |
| Factor #3 Workplace Support WITH | | | |
| Training F#1 | -0.199*** | 0.038 | < .001 |
| Connectedness F#2 | 0.193*** | 0.04 | < .001 |
| FACTOR #4 Skill and Knowledge to Support Parent and their Parenting WITH | | | |
| Training F#1 | 0.063 | 0.039 | 0.1 |
| Connectedness F#2 | 0.565*** | 0.031 | < .001 |
| Workplace Support F#3 | 0.207*** | 0.036 | < .001 |
| FACTOR #5 Referral WITH | | | |
| Training F#1 | 0.182*** | 0.047 | < .001 |
| Connectedness F#2 | 0.459*** | 0.047 | < .001 |
| Workplace Support F#3 | 0.343*** | 0.043 | < .001 |
| Parenting Support #4 | 0.338*** | 0.038 | < .001 |
| FACTOR #6 Psycho Education WITH | | | |
| Training F#1 | -0.094* | 0.046 | 0.04 |
| Connectedness F#2 | 0.275*** | 0.048 | < .001 |
| Workplace Support F#3 | 0.511*** | 0.038 | < .001 |
| Parenting Support F#4 | 0.263*** | 0.042 | < .001 |
| Referral F#5 | 0.564*** | 0.044 | < .001 |
| FACTOR #7 Understanding The Family Model WITH | | | |
| Training F#1 | -0.322*** | 0.038 | < .001 |
| Connectedness F#2 | 0.186*** | 0.048 | < .001 |
| Workplace Support F#3 | 0.344*** | 0.038 | < .001 |
| Parenting Support F#4 | 0.286*** | 0.038 | < .001 |
| Referral F#5 | 0.330*** | 0.050 | < .001 |
| Psycho Education F#6 | 0.444*** | 0.040 | < .001 |
| FACTOR #8 Skills and Knowledge to Support Children WITH | | | |
| Training F#1 | -0.007 | 0.042 | 0.875 |
| Connectedness F#2 | 0.491*** | 0.038 | < .001 |
| Workplace Support F#3 | 0.285*** | 0.035 | < .001 |
| Parenting Support F#4 | 0.406*** | 0.031 | < .001 |
| Referral F#5 | 0.520*** | 0.039 | < .001 |
| Psycho Education F#6 | 0.384*** | 0.042 | < .001 |
| Understanding The Family Model #7 | 0.273*** | 0.039 | < .001 |
| FACTOR #9 Child Protection WITH | | | |
| Training F#1 | 0.249*** | 0.041 | < .001 |
| Connectedness F#2 | 0.370*** | 0.044 | < .001 |
| Workplace Support F#3 | 0.183*** | 0.043 | < .001 |
| Parenting Support F#4 | 0.484*** | 0.035 | < .001 |
| Referral F#5 | 0.236*** | 0.042 | < .001 |
| Psycho Education F#6 | 0.054 | 0.049 | 0.265 |
| Understanding The Family Model F#7 | 0.131** | 0.043 | 0.003 |
| Skills and Knowledge to Support Children F#8 | 0.246 | 0.045 | < .001 |
| FACTOR #10 Time and Workload WITH | | | |
| Training F#1 | -0.116** | 0.04 | 0.004 |
| Connectedness F#2 | 0.137** | 0.04 | 0.001 |

*(Continued)*

**Table 5.** (Continued)

| | Correlations | Std Error | *p* value |
|---|---|---|---|
| Workplace Support F#3 | 0.473*** | 0.032 | < .001 |
| Parenting Support F#4 | 0.074 | 0.039 | 0.058 |
| Referral F#5 | 0.296*** | 0.043 | < .001 |
| Psycho Education F#6 | 0.381*** | 0.041 | < .001 |
| Understanding The Family Model F#7 | 0.147*** | 0.042 | < .001 |
| Skills and Knowledge to Support Children F#8 | 0.348*** | 0.037 | < .001 |
| Child Protection F#9 | 0.005 | 0.045 | 0.913 |
| FACTOR #11 Service Availability WITH | | | |
| Training F#1 | -0.321*** | 0.041 | < .001 |
| Connectedness F#2 | 0.062 | 0.048 | 0.192 |
| Workplace Support F#3 | 0.412*** | 0.039 | < .001 |
| Parenting Support F#4 | 0.137** | 0.043 | 0.001 |
| Referral F#5 | 0.31*** | 0.047 | < .001 |
| Psycho Education F#6 | 0.246*** | 0.049 | < .001 |
| Understanding The Family Model F#7 | 0.149** | 0.047 | 0.002 |
| Skills and Knowledge to Support Children F#8 | 0.162*** | 0.045 | < .001 |
| Child Protection F#9 | 0.044 | 0.049 | 0.362 |
| Time and Workload #10 | 0.495*** | 0.037 | < .001 |
| FACTOR #12 Engagement Issues WITH | | | |
| Training F#1 | -0.058 | 0.049 | 0.234 |
| Connectedness F#2 | 0.245*** | 0.049 | < .001 |
| Workplace Support F#3 | 0.268*** | 0.042 | < .001 |
| Parenting Support F#4 | 0.278*** | 0.04 | < .001 |
| Referral F#5 | 0.295*** | 0.052 | < .001 |
| Psycho Education F#6 | 0.178*** | 0.051 | < .001 |
| Understanding The Family Model F#7 | 0.132** | 0.048 | 0.006 |
| Skills and Knowledge to Support Children F#8 | 0.563*** | 0.036 | < .001 |
| Child Protection F#9 | 0.107* | 0.051 | 0.036 |
| Time and Workload #10 | 0.443*** | 0.04 | < .001 |
| Service Availability #11 | 0.271*** | 0.053 | < .001 |

influenced by both the individual professional and their core overlapping practices as well as the organizational structures which surround them.

Following robust statistical analysis including EFA and CFA, results suggest that 12 sub-scales containing a total of 46 items was most optimal using the current sample. These sub-scales are slightly different from the original scale with five related to the individual professional (i.e. knowledge and skill, willingness to undertake training), three describing family focused behaviours (i.e. referral, psycho education and skill and knowledge to support the parent and parenting), and four organizational factors impacting these behaviours, such as time and workload and workplace support.

Test of internal consistency using Cronbach's alpha suggest that the 46-item version is reliable (.85) with subscale scores ranging from .60 to .83 with the exception of Psycho Education (.48) and Child protection (.49), each of which had 3 items. As Ueno et al [38] note "In some literatures it is reported that reliability is "acceptable" if Cronbach's alpha is greater than .60 or .70 but it is recommended that care should be taken when using these subscales" (p.66). Our findings also coincide with that of other studies which found that while the FFMHPQ had

good validity and was reasonably reliable, internal reliability was poorer within some subscales [33, 37–39, 46]. Therefore, like existing research we feel it imperative for future research using the FFMHPQ in the context of multiple service/ sector assessment, to address and improve the reliability of the weaker items.

Nonetheless, the 12 subscales identified are meaningful and consistent with substantive theories and proposed models of FFP and with our objective, which suggest that specific family focus practises are influenced by both the individual professional and their core overlapping practices as well as the organisational structures which surround them [22, 23, 43, 44]. For example, we found inter-correlations that are consistent with known professional and organizational processes identified as either promoting or hindering FFP [8, 12, 38] such as *Time and workload* with *Skills and knowledge to support children*, and *Referral* with *Workplace support*. We also observed a negative correlation between *Training and Understanding The Family Model* and *Training and Service Availability*. We suggest this may reflect a relationship between those indicating a need/willingness for further family focused training, and those who indicated they had less understanding of The Family Model and that those who felt they needed more training also perceived a lack of services to refer parents and children to. Such findings are perhaps similar to Ueno et al. [38] who highlight how professionals who had previous training in FFP had more confidence to engage in FFP and indicated that they undertook more family focused activities including referral of parents and children to other services than those professionals without training.

Moreover, current findings also support behavioural intention models such as the Theory of Planned Behaviour which suggests that an individual's attitudes and knowledge may influence their behaviour [47]. More broadly, our findings coincide with the wider literature on FFP which suggests that if professionals have positive attitudes towards FFP and the necessary knowledge and skills to practice this way, they are more likely to adopt a whole of family approach [8, 12, 48]. This connection between attitudes, knowledge and skill and behaviour highlights the importance of organisations having effective implementation strategies to embed FFP [36, 45]. Future research should further examine these relationships and identify key predictors of FFP within adult mental health and children's services.

The results of the current study are also similar to that of Maybery et al. [30] who identified 14 factors (including professional and organizational) when used in adult mental health services in Australia and Ueno et al. [38] who identified 13 factors in adult mental health services in Japan. The meaningful clusters and similarity of results, for the most part, between the three countries and cultures in terms of the dimensions identified suggests that the scale is a good measure of family focused behaviours and factors that impact these. Differences between the current results and that of the original measure [30], and an adaption by Ueno et al. [38], relate mostly to labelling of factors and clustering of items within. For instance, while Maybery et al. [30] identified two separate subscales related to workplace support and professional development, items in the current study loaded into one factor labelled "workplace support" which had good reliability (.81). The differences between results concerning this scale may be explained by contextual factors that pertain to the pathways offered to professionals to engage in FFP. The results of the current study and that of Maybery et al. [30] and Ueno et al. [38], differ substantially to that of Leonard et al. [37] who in a population of health visitors identified two factors related to the individual professional and organisation. These differences suggest that while the measure is flexible it may have more relevance for some services, sectors and professionals than others, depending on the professionals' remit, understanding and operalisation of FFP. This needs consideration in future research examining FFP in different services other than adult mental health and children's services.

In adapting the measure for the current study, the authors, as previously noted, also developed three additional subscales, one of which aimed to measure professionals' knowledge and skills in supporting parents to cope with the impact of parenting on their mental health. While Maybery et al. [30] identified a single subscale on knowledge and skills, (primarily related to supporting children), in the current study these items loaded onto two subscales, measuring knowledge and skills to support either parents or children. Two distinct factors related to knowledge and skill within the current study make sense; they reflect the additional items generated and distinction between knowledge and skills to support children and to support parents as discussed in the literature. A key barrier cited in the literature for FFP is the lack of an integrated approach to service provision and lack of interagency cooperation between adult mental health and children's services [24, 35]. The predominant focus of either service on parents or children as opposed to both together has led to professionals in adult mental health services perceiving that they do not have the skills to support children and for professionals in children's services to support parents. Our findings further underscore this distinction in knowledge, skills and practice between services and sectors and this should be factored into future benchmarks and organizational support of FFP.

The current study also identified two additional individual professional factors, related to knowledge and skills, including understanding of [17, 27] The Family Model and child protection protocols. As previously noted, professionals' understanding and application of The Family Model is particularly important in NI considering the HSCB has endorsed its use since 2009 as a framework to embed FFP within services and to structure provision of in service training in this area [8, 28]. The inclusion of this new and reliable subscale on TFM will provide future scope for those organisations who have endorsed this model to monitor and improve its translation in practice. A new subscale on child protection was also developed because while the subscale on connectedness, first identified by Maybery et al. [30], measures professionals' skill and knowledge capacity beyond assessing abuse and neglect, the original measure did not include items measuring understanding of child protection protocols. It is important to measure professionals' understanding of child protection protocols in NI and elsewhere because they can be used to inform and encourage an early intervention approach to support parents in their parenting as opposed to being solely applied in response to situations where children's well-being is at risk [14]. Grant et al. [14] found that HSC professionals perceived that they were better able to engage parents around parenting when they highlighted their responsibilities in relation to child protection while at the same time emphasizing the importance of intervening early to support both parents and children and to keeping families together when possible.

## Limitations

Although the professional composition of respondents was similar to those in other studies [30, 38], the sample was drawn from two different sectors making it difficult to directly compare findings with previous research. The FFMHPQ is a self-report questionnaire from the professionals' perspectives and this may not be a fully accurate reflection of actual practice. It may also differ from perspectives of service users, families and managers. Observational research might be conducted to provide additional data regarding the dimensions of FFP and could include the perspectives of family members and managers. While the FFMHPQ had documented validity and reliability in the Australian context [30], in the current study two of the subscales, psychoeducation and child protection, had Cronbach's alpha coefficients below 0.60, hence care should be taken when using these. In addition, in future research, efforts could be made to improve their reliability. The nature and specificity of the items within these

two subscales could be revised in order to better reflect professional's experience in relation to the concepts being measured.

## Conclusion

In conclusion, this study identifies a 12-subscale measure of professionals' FFP in adult mental health and children's services. The FFMHPQ (NI version) was adapted from the original measure [30], in conjunction with the developers of the instrument in response to the emerging literature on PMI, FFP and organisational developments in FFP in NI. This study has developed further understanding of the dimensions measured within the FFMHPQ and consistency of findings with previous studies. Indeed, the factors identified in our CFA indicate attitudes and behaviours that are known to play a key role in the propensity to engage with FFP. They also represent key contextual factors that can affect the former ones. The associations between factors displayed in our results support the conclusion that the FFMHPQ measure provides a reliable description of FFP engagement and its supporting factors. The test of internal consistency using Cronbach's alpha suggest that the 46-item version is reliable (.85) with subscale scores, aside from two, ranging from.60 to.83. Furthermore, the 12-factor model provided good fit to the data, based on common indices used in CFA.

Thus, the results of this study indicate the FFMHPQ is a reliable tool that addresses the paucity of measurement tools in this area, which persists despite increasing recommendations for professionals to engage in FFP and for organizations to benchmark and promote it. Our results indicate that the FFMHPQ can be used to help identify professionals' key formative needs in relation to FFP as well as what predicts it; thereby presenting a means of benchmarking, monitoring and evaluating service provision and further developing training programmes. Further research using our adapted version of the FFMHPQ should examine and attempt to replicate the psychometric properties of the measure as well as extend its component structure to any items or factors not included in the current research.

## Supporting information

**S1 Table. Correlations matrix of 14 factor solution.**
(DOCX)

**S2 Table. Items factor loadings (Est. = Estimate), Standard Errors (S.E.) and z of items by factors in the 12-factor solution.**
(DOCX)

**S3 Table. Standardised loadings, standard errors, and p values of item loadings by items and by factors.**
(DOCX)

**S1 File.**
(DOCX)

**S1 Data.**
(OUT)

**S2 Data.**
(OUT)

## Acknowledgments

We thank the professionals who completed the survey and the staff in the participating services for their assistance with recruitment and logistics.

## Author Contributions

**Conceptualization:** Anne Grant, Susan Lagdon, Gavin Davidson.

**Data curation:** Anne Grant, Susan Lagdon, John Devaney, Gavin Davidson, Joe Duffy, Oliver Perra.

**Formal analysis:** Anne Grant, Susan Lagdon, Oliver Perra.

**Funding acquisition:** Anne Grant.

**Investigation:** Anne Grant, Susan Lagdon, John Devaney.

**Methodology:** Anne Grant, Susan Lagdon, John Devaney, Gavin Davidson, Oliver Perra.

**Project administration:** Anne Grant, Susan Lagdon.

**Resources:** Anne Grant, Susan Lagdon.

**Software:** Anne Grant, Susan Lagdon, Oliver Perra.

**Supervision:** Anne Grant.

**Validation:** Anne Grant, Susan Lagdon, Oliver Perra.

**Visualization:** Anne Grant, Susan Lagdon, Oliver Perra.

**Writing – original draft:** Anne Grant, Susan Lagdon, Oliver Perra.

**Writing – review & editing:** Anne Grant, Susan Lagdon, John Devaney, Gavin Davidson, Joe Duffy, Oliver Perra.

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
