## [Decision Letter · Decision Letter 0]

31 Jan 2022

PONE-D-21-34357Validation of the Family Focused Mental Health Practice Questionnaire in measuring Health and Social Care Professionals’ Family Focused PracticePLOS ONE

Dear Dr. Grant Anne,

Thank you for submitting your manuscript to PLOS ONE. After careful consideration, we feel that it has merit but does not fully meet PLOS ONE’s publication criteria as it currently stands. Therefore, we invite you to submit a revised version of the manuscript that addresses the points raised during the review process.

We look forward to receiving your revised manuscript.

Kind regards,

Shelina Bhamani, PhD, Public health

Academic Editor

PLOS ONE

Journal Requirements:

3. We note that Table 2 may include questionnaire items that may have been previously published. The reproduction of previously published work has implications for the copyright that may apply to these publications. We would be grateful if you could clarify whether you have obtained permission from the original copyright holder to republish these items under a CC BY license. If you have not obtained permission to publish these items please remove them from your manuscript. You may wish to replace the text you have removed with relevant question numbers/ brief descriptions of each item; please be sure to include any relevant references and in-text citations.

Additional Editor Comments :

Please see the comments attached by the reviewers and make changes.

Reviewer 1:

The manuscript has sound piece of scientific research with data that supports the conclusions.

Statistical analysis been performed appropriately.

The manuscript is presented in an intelligible fashion and written in standard English.

Reviewer 2:

The manuscript mentioned that the FFMHPQ was refined within the Irish context (page 9) but what are the changes made were not discussed appropriately.

The reason for choosing the sites was also not discussed.

What was the inclusion and exclusion criteria of selecting HSC professionals needs to elaborate.

Generalizability of the study needs to discusses both; at the national and global level.

Reviewers' comments:

Reviewer's Responses to Questions

**Comments to the Author**

1. Is the manuscript technically sound, and do the data support the conclusions?

Reviewer #1: Yes

Reviewer #2: Yes

2. Has the statistical analysis been performed appropriately and rigorously? 

Reviewer #1: Yes

Reviewer #2: Yes

3. Have the authors made all data underlying the findings in their manuscript fully available?

Reviewer #1: Yes

Reviewer #2: No

4. Is the manuscript presented in an intelligible fashion and written in standard English?

Reviewer #1: Yes

Reviewer #2: Yes

5. Review Comments to the Author

Reviewer #1: The manuscript has sound piece of scientific research with data that supports the conclusions.

Statistical analysis been performed appropriately.

The manuscript is presented in an intelligible fashion and written in standard English.

Reviewer #2: The manuscript mentioned that the FFMHPQ was refined within the Irish context (page 9) but what are the changes made were not discussed appropriately.

The reason for choosing the sites was also not discussed.

What was the inclusion and exclusion criteria of selecting HSC professionals needs to elaborate.

Generalizability of the study needs to discusses both; at the national and global level.

6. PLOS authors have the option to publish the peer review history of their article (what does this mean?). If published, this will include your full peer review and any attached files.

Reviewer #1: No

Reviewer #2: **Yes:**

---

## [Author Response · Author response to Decision Letter 0]

3 Feb 2022

please see response in table attached entitled response to reviewers.

---

## [Editor Report · Decision Letter 1]

3 Aug 2022

 PLOS ONE

Dear,

Thank you for submitting your manuscript to PLOS ONE. After careful consideration, we feel that it has merit but does not fully meet PLOS ONE’s publication criteria as it currently stands. Therefore, we invite you to submit a revised version of the manuscript that addresses the points raised during the review process.

A marked-up copy of your manuscript that highlights changes made to the original version. You should upload this as a separate file labeled 'Revised Manuscript with Track Changes'.An unmarked version of your revised paper without tracked changes. You should upload this as a separate file labeled 'Manuscript'.If applicable, we recommend that you deposit your laboratory protocols in protocols.io to enhance the reproducibility of your results. Protocols.io assigns your protocol its own identifier (DOI) so that it can be cited independently in the future. For instructions see: https://journals.plos.org/plosone/s/submission-guidelines#loc-laboratory-protocols. Additionally, PLOS ONE offers an option for publishing peer-reviewed Lab Protocol articles, which describe protocols hosted on protocols.io. Read more information on sharing protocols at https://plos.org/protocols?utm_medium=editorial-email&utm_source=authorletters&utm_campaign=protocols.

We look forward to receiving your revised manuscript.

Kind regards,

Shelina Bhamani, PhD

Academic Editor

PLOS ONE

Journal Requirements:

Additional Editor Comments (if provided):

Dear, There are several language and grammar mistakes. Please proofread the documents before proceed for next step. Some of the sentence use is not clear for example "in the current study two of the subscales, psychoeducation and child protection, had Cronbach’s alpha coefficients below 0.60, hence care should be taken when using these" Please write complete statements with clear indication. Please add one heading on Theoretical foundation of Family Focused Mental Health Practice questionnaire. Please include more literature review and background of study. Please add future recommendation and future direction of your study

Reviewers' comments:

Please improve the language of manuscript and please provide more details in table on how questions were developed and provide literature background.

---

## [Author Response · Author response to Decision Letter 1]

13 Oct 2022

Dear editor please see our response detailed in:

revised manuscript with track and without track changes

rebuttal letter with our response to your constructive suggestions.

Thanks

---

## [Editor Report · Decision Letter 2]

19 Dec 2022

PONE-D-21-34357R2Validation of the Family Focused Mental Health Practice Questionnaire in measuring Health and Social Care Professionals’ Family Focused PracticePLOS ONE

Dear Dr. Grant Anne,

Thank you for submitting your manuscript to PLOS ONE. After careful consideration, we feel that it has merit but does not fully meet PLOS ONE’s publication criteria as it currently stands. Therefore, we invite you to submit a revised version of the manuscript that addresses the points raised during the review process.

1- Please provide strobe statement and Strobe checklist for cross-sectional studies supplementary data.

2- Please provide your Raw data as supplementary file. 

3- Please provide Family Focused Mental Health Practice Questionnaire adapted as supplementary file.

4- You did not indicate your hypothesis in the present study. There is a need of give hypothesis in the introduction and then compare it with the output in the discussion and i wonder that you have mentioned inside your discussion that  "We also observed a negative correlation between Training and Understanding The Family Model and Training and Service Availability" But we do not have your hypothesis, which you were testing initially. These are standard information that is available in manuscript. 

5-Please explain more in details on Cronbach’s alpha coefficients below 0.60 and justify in details and explain the imperative approach for future research using the FFMHPQ.

5-Please check the language and grammar again. Did you proof read the documents?

We look forward to receiving your revised manuscript.

Kind regards,

Muhammad Shahzad Aslam, Ph.D.,M.Phil., Pharm-D

Academic Editor

PLOS ONE

Journal Requirements:

Additional Editor Comments:

1- Please provide strobe statement and Strobe checklist for cross-sectional studies supplementary data.

2- Please provide your Raw data as supplementary file.

3- Please provide Family Focused Mental Health Practice Questionnaire adapted as supplementary file.

4- You did not indicate your hypothesis in the present study. There is a need of give hypothesis in the introduction and then compare it with the output in the discussion and i wonder that you have mentioned inside your discussion that "We also observed a negative correlation between Training and Understanding The Family Model and Training and Service Availability" But we do not have your hypothesis, which you were testing initially. These are standard information that is available in manuscript.

5-Please explain more in details on Cronbach’s alpha coefficients below 0.60 and justify in details and explain the imperative approach for future research using the FFMHPQ.

5-Please check the language and grammar again. Did you proof read the documents?

---

## [Author Response · Author response to Decision Letter 2]

6 Apr 2023

We have further responded to reviewers constructive feedback and have highlighted changes using track changes and uploaded an additional response table and additional files including EFA & CFA output, strobe checklist and copy of survey.

We have removed figure from manuscript and we have removed funding statement also.

---

## [Editor Report · Decision Letter 3]

3 May 2023

Validation of the Family Focused Mental Health Practice Questionnaire in measuring Health and Social Care Professionals’ Family Focused Practice

PONE-D-21-34357R3

Dear,

We’re pleased to inform you that your manuscript has been judged scientifically suitable for publication and will be formally accepted for publication once it meets all outstanding technical requirements.

Kind regards,

Muhammad Shahzad Aslam, Ph.D.,M.Phil., Pharm-D

Academic Editor

PLOS ONE
---

## [Editor Report · Acceptance letter]

12 May 2023

PONE-D-21-34357R3 

Validation of the family focused mental health practice questionnaire in measuring health and social care professionals’ family focused practice 

Dear Dr. Grant:

I'm pleased to inform you that your manuscript has been deemed suitable for publication in PLOS ONE. Congratulations! Your manuscript is now with our production department. 

Kind regards, 

on behalf of

Dr. Muhammad Shahzad Aslam 

Academic Editor

PLOS ONE